# Involvement of *Fusobacterium* Species in Oral Cancer Progression: A Literature Review Including Other Types of Cancer

**DOI:** 10.3390/ijms21176207

**Published:** 2020-08-27

**Authors:** Natsumi Fujiwara, Naoya Kitamura, Kaya Yoshida, Tetsuya Yamamoto, Kazumi Ozaki, Yasusei Kudo

**Affiliations:** 1Department of Oral Health Care Promotion, Tokushima University Graduate School of Biomedical Sciences, 3-18-15 Kuramoto, Tokushima 770-8504, Japan; nfujiwara@tokushima-u.ac.jp (N.F.); ozaki@tokushima-u.ac.jp (K.O.); 2Department of Oral Biology & Diagnostic Sciences, The Dental College of Georgia, Augusta University, 1120 15th Street, Augusta, GA 30912, USA; 3Department of Oral and Maxillofacial Surgery, Kochi Medical School, Kochi University, Kohasu, Oko-cho, Nankoku 783-8505, Japan; nkitamura@kochi-u.ac.jp (N.K.); yamamott@kochi-u.ac.jp (T.Y.); 4Department of Oral Health Care Education, Tokushima University Graduate School of Biomedical Sciences, 3-18-15 Kuramoto, Tokushima 770-8504, Japan; kaya@tokushima-u.ac.jp; 5Department of Oral Bioscience, Tokushima University Graduate School of Biomedical Sciences, 3-18-15 Kuramoto, Tokushima 770-8504, Japan

**Keywords:** *Fusobacterium nucleatum*, oral cancer, cancer development and progression

## Abstract

Chronic inflammation caused by infections has been suggested to be one of the most important cause of cancers. It has recently been shown that there is correlation between intestinal bacteria and cancer development including metastasis. As over 700 bacterial species exist in an oral cavity, it has been concerning that bacterial infection may cause oral cancer. However, the role of bacteria regarding tumorigenesis of oral cancer remains unclear. Several papers have shown that *Fusobacterium* species deriving the oral cavities, especially, play a crucial role for the development of colorectal and esophageal cancer. *F. nucleatum* is a well-known oral bacterium involved in formation of typical dental plaque on human teeth and causing periodontal diseases. The greatest characteristic of *F. nucleatum* is its ability to adhere to various bacteria and host cells. Interestingly, *F. nucleatum* is frequently detected in oral cancer tissues. Moreover, detection of *F. nucleatum* is correlated with the clinical stage of oral cancer. Although the detailed mechanism is still unclear, *Fusobacterium* species have been suggested to be associated with cell adhesion, tumorigenesis, epithelial-to-mesenchymal transition, inflammasomes, cell cycle, etc. in oral cancer. In this review, we introduce the reports focused on the association of *Fusobacterium* species with cancer development and progression including oral, esophageal, and colon cancers.

## 1. Introduction

Oral cancer, predominantly oral squamous cell carcinoma (OSCC), is a significant health problem and is regarded as the main cause of death from oral diseases in many countries. Traditional risk factors of oral cancer include alcohol abuse, tobacco and tobacco-derivate chewing, and oral virus infections. Other factors include infections, exposure to ionizing radiation, and environmental pollutants [1]. Thus, various causes of cancer are known to be closely involved in lifestyle choices, such as smoking, drinking, and diet.

Chronic inflammation caused by infections has been suggested to be one of the most important cause of cancers [2]. Indeed, infection of various viruses, bacteria, and parasites, such as Hepatitis B virus (HBV), Hepatitis B virus (HCV), Epstein–Barr virus (EBV), human papilloma virus (HPV), human herpes virus 8 (HHV8), human thymus-derived-cell leukemia/lymphoma virus-1 (HTLV-1), human immunodeficiency virus (HIV), *Helicobacter pylori*, Schistosomiasis, and liver flukes are well-known causes of cancer [2]. Recently, commensal bacteria are also involved in carcinogenesis. A huge number of different commensal bacteria naturally colonize in the human body in a relatively stable equilibrium. Microbial-immune network correlates gut bacteria with the whole-body health, and the failure of immune homeostasis manifests significant impact on various diseases, which might result in cancer [3,4,5]. Indeed, some commensal bacteria including *Peptostreptococcus anaerobius* [6], enterotoxigenic *Bacteriodes fragiles* [7], and *Escherichia coli* [8] are involved in carcinogenesis of colorectal cancer (CRC).

In the oral cavity, bacteria, fungi, viruses, and archaea naturally colonize in different habitats including the teeth, gingival sulcus, tongue, cheeks, hard and soft palates, and tonsils. The oral microbiota refers to a highly varied and complicated ecosystem of these organisms. Over 700 bacterial species are endemic to the oral cavity, and indigenous oral flora act to prevent the settlement of foreign bacteria. Some bacteria of the oral cavity are harmful and can cause serious disease, while many of the oral bacteria are in fact beneficial in preventing diseases. Thus, the oral cavity is inhabited by complex multispecies bacterial communities that usually exist in a balanced immunoinflammatory state with the host [9]. It is now established that many chronic inflammatory conditions are caused by an imbalance between host–microbiota interactions, resulting in a dysbiotic community, deregulated immune responses, and eventually disease outcomes [10]. Oral commensal bacteria play a critical role in the development of oral diseases, including periodontal disease and tooth loss, and maintenance of a normal oral physiological environment [11,12]. Moreover, oral commensal bacteria are known to be involved in the pathogenesis and development of systematic diseases, such as pneumonia, cardiovascular diseases, diabetes, dementia, etc. It has been of concern that oral commensal bacteria may be involved in the pathogenesis of OSCC [13,14]. However, it remains unclear the role of bacteria regarding carcinogenesis of OSCC, even though a lot of bacteria inhabit in the oral cavity. Interestingly, it recently has been shown that one of oral commensal bacteria, *Fusobacterium* species, especially, play a crucial role for the development of CRC [15,16,17]. As *Fusobacterium* species are commensal bacteria in the oral cavity, the cumulative evidences of *Fusobacterium* species in CRC make us hypothesize that *Fusobacterium* species may be involved in the pathogenesis and development of OSCC. In this review, we introduce the reports focused on the association of *Fusobacterium* species with cancer development and progression.

## 2. *Fusobacterium* Species

The genus *Fusobacterium* are Gram-negative, non-spore forming, non-motile, pleomorphic, and obligate anaerobic bacilli, which inhabit the human oral cavity, oropharynx, upper respiratory, gastrointestinal, and female genitourinary tracts as part of the normal flora [18]. Nine species including *F. nucleatum* (subspecies *nucleatum*, *polymorphum*, *vincentii*, *animalis*, *fusiforme*, and *canifelium*), *F. necrophorum* (subspecies *necrophorum* and *funduliforme*), *F. ulcerans*, *F. gonidiaformans*, *F. mortiferum*, *F. naviforme*, *F. necrogenes*, *F. russii*, and *F. varium* cause human infections of the head and neck, chest, lung, liver, and abdomen [19]. *F. nucleatum* is a well-known oral bacterium that forms typical dental plaque on human teeth and is involved in periodontal diseases [20]. The greatest characteristic of *F. nucleatum* is to adhere to various bacteria and cells. *F. nucleatum* is a central species in physical interactions between Gram-positive and Gram-negative species that are likely to be important for biofilm formation, and it is considered as a bridge for the attachment of commensals that colonize the tooth and epithelial surface with true pathogens [21,22]. *F. nucleatum* has two types of substance—fimbriae and non-fimbrial adhesin—for attaching to other bacteria and cells. These substances contribute to colonization and also bacterial pathogenesis and infection. *Fusobacterium* adhesion A (FadA), a fimbrial adhesin protein, was recently shown to be required for bacterial attachment and invasion of gingival epithelial and endothelial cells [23]. FadA has two forms: pre-FadA and mature FadA (mFadA; 111 amino acids). The pre-FadA (129 amino acids) is a non-secreted protein and is associated with the inner membrane. The mFadA is a secreted protein and is easily dissociated from bacteria. Mixtures of pre-FadA and mFadA form high-molecular-weight aggregates, which are required for attachment and invasion of host cells [24]. FadA is highly conserved among oral fusobacteria, such as *F. nucleatum*, *F. periodonticum* and *F. simiae*, whereas it is absent in non-oral fusobacterial species, including *F. gonidiaformans*, *F. mortiferum*, *F. naviforme*, *F. russii*, and *F. ulcerans* [23].

Two distinct types of adherence protein have been found in fusobacteria and classified based on their inhibition mode using either D-galactose or L-arginine [25]. Adherence to predominantly Gram-negative late-colonizing bacteria, including *Porphyromonas gingivalis*, *Tannerella forsythia*, and *Treponema denticola*, is associated with galactose-inhibiting (lectin-like) interactions, while adherence to mainly Gram-positive early colonizing species is associated with arginine-inhibiting (nonlectin-like) interactions. Three components, a 40–42 kDa major outer membrane porin protein (FomA), and 39.5-kDa and 30-kDa polypeptides, have been suggested as possible adhesins involved in the lectin-like interbacterial co-aggregation of *F. nucleatum* [26,27,28]. These adherent factors of *Fusobacterium* play important roles in tumor progression as discussed in the next section.

## 3. *F. nucleatum* and Cancer

### 3.1. F. nucleatum in Colorectal Cancer

Elevated *F. nucleatum* levels were significantly detected in colon tissue of colon cancer patients compared with healthy people and *F. nucleatum* inhabited on colorectal tissues in a half of colon cancer cases [29,30], suggesting that *F. nucleatum* may be involved in CRC development. Moreover, *F. nucleatum* colonies in primary CRC tissues were also identified in the metastatic liver tissues [31]. Komiya et al. reported that *F. nucleatum* was detected in both CRC and oral saliva samples, 75% of which was the same strain [32]. Cumulative clinical evidences suggested that the enrichment of *F. nucleatum* might be related to CRC metastasis [29,31,33,34,35]. Moreover, *F. nucleatum* was associated with CRC recurrence and resistance to chemotherapy by activating the autophagy pathway [36]. Thus, *F. nucleatum* that may be derived from oral cavity is frequently detected in CRC samples and is suggested to be involved in malignant behaviors of CRC.

As described above, several reports show the association of *F. nucleatum* with carcinogenesis or metastasis of CRC. Kostic et al. reported that intake of *F. nucleatum* to Apc^MIN/+^ mice, which causes gastrointestinal cancer, promoted small and large intestinal tumorigenesis [37]. Moreover, *F. nucleatum* was enriched in tumor tissues compared with *Streptococcus* species [37]. Importantly, abundance of *Fusobacterium* was frequently detected in the stool of patients as well as in the adenomas and CRC tissues [37]. In the intestinal tumor microenvironment, *F. nucleatum* selectively expanded myeloid-derived immune cells but not lymphoid immune cells, leading to tumor progression [37]. Moreover, *F. nucleatum* promotes metastasis of CRC by modulating *KRT7-AS*/*KRT7* [38].

The outer membrane proteins of *F. nucleatum*, such as FadA and Fibroblast activation protein 2 (Fap2) have been focused as CRC pathogen. E-cadherin acts as a tumor suppressor by modulating β-catenin pathway [39,40]. Loss or heterogeneous expression of E-cadherin correlated with advanced stages and poor prognosis of CRC patients [41]. Rubinstein et al. demonstrated that *F. nucleatum* attached and invaded into CRC cells via the FadA-mediated interaction with E-cadherin [15]. Moreover, FadA promotes E-cadherin-mediated CRC tumor growth in xenograft mice via NF-κB activation, producing proinflammatory cytokines such as IL-6, IL-8, and IL-18 and upregulation of Wnt signaling gene (Wnt-7a, -7b, and -9a), Myc, and Cyclin D1 [15]. Interestingly, patients with adenoma and adenocarcinoma in colon had elevated *FadA* gene expression 10–100 times higher compared to normal individuals [15].

Fap2 also plays a critical function in the development of CRC. Fap2, a 390-KDa protein encoded by the *Fap2* gene of *F. nucleatum*. Fap2 is suggested to be a galactose-sensitive hemagglutinin and adhesin that is likely to play a role in the virulence of fusobacteria [42]. Fap2 inhibits the immune cells, such as T cells and NK cells for killing the CRC cells via a immunoreceptor, TIGIT [16]. Moreover, acetylgalactosamine (Gal-GalNAc) is recognized by fusobacterial Fap2, which functions as a Gal-Gal-NAc lectin [17]. Intravenously injected *F. nucleatum* localized to mouse CRC tissues in a Fap2-dependent manner, suggesting that fusobacteria use a hematogenous route to reach colon adenocarcinomas via interaction between Fap2 and Gal-GalNAc [17]. Interestingly, Gal-GalNAc is highly expressed in adenocarcinomas including ovary, stomach, colon and lung [43,44]. Although it is still unclear the role of GalNAc in OSCC, N-acetylgalactosaminyltransferase 3 (GalNAc-T3), which regulates the critical initial steps of mucin-type O-glycosylation, may play a role in the pathogenesis of recurrence of early stage of OSCC [45]. These observations suggest that Fap2 of *Fusobacterium* may modulate tumor progression by binding to Gal-GalNAc on cancer cells.

Today, there are several therapeutic approaches available with which to modify gut microbiota on CRC management. To alter the composition and the activity of gut microbiota, several therapeutic methods, such as fecal microbiota transplantation and administration of prebiotics, probiotics, and synbiotics, are available for CRC patients [46]. In addition to these methods, oral hygiene managements for reducing the amount of *F. nucleatum* may contribute to the prevention of CRC.

### 3.2. F. nucleatum in Esophageal Cancer

The involvement of *F. nucleatum* in the development of esophageal cancer has also been reported. Yamamura et al. reported that DNA levels of *F. nucleatum* in esophageal cancer tissues were higher than those in pared adjacent non-tumor tissues [47]. Moreover, esophageal squamous cell carcinoma (ESCC) tissues contained more *Fusobacterium* (3.2% vs. 1.3%) and less *Streptococcus* (12.0% vs. 30.2%), compared to the non-tumor tissues [48]. Importantly, higher DNA levels of *F. nucleatum* in ESCC patients were significantly associated with cancer-specific survival [47], predicted poor recurrence-free survival, and poor response to neoadjuvant chemotherapy [49]. In addition, C-C Motif Chemokine Ligand 20 (CCL20), which plays crucial roles in cell proliferation and migration, is significantly upregulated in *F. nucleatum*-positive esophageal cancer tissues [47]. Thus, *F. nucleatum* is involved in the development of colorectal and esophageal cancers.

### 3.3. Fusobacterium Species and Oral Cancer

There are many reports showing that *Fusobacterium* were detected in OSCC tissues [50,51]. The ratio between aerobes and anaerobes within the biofilms on the surfaces of OSCC tissues was approximately 1:2, whereas that in the healthy control was 2:1, indicating that OSCC surfaces provide an important reservoir for anaerobic bacteria [52]. *F. nucleatum* subsp. *polymorphum* was the most significantly overrepresented species in OSCC compared to the control group with deep-epithelium swabs [53]. Perera et al. also reported that the *Fusobacterium* was enriched in OSCC biopsy compared to fibroepithelial polyp as a control [54]. Moreover, significantly greater bacterial diversity was observed in the swab of OSCC sites than that of normal sites [55]. Thus, distribution of bacteria including *Fusobacterium* in OSCC tissues may be distinct from that in healthy oral mucosal tissues. Yang et al. reported that *F. periodonticum* and several bacteria in oral rinse samples were associated with OSCC, and they progressively increased in abundance from stage I to stage IV [56].

*Fusobacterium* can invade into epithelial cells. Compared to *P. gingivalis*, wild-type *F. nucleatum* 12,230 remarkably adheres and invades into human gingival epithelial cells (HGECs) [57]. Interestingly, the spontaneous mutant of *F. nucleatum* was unable to invade into HGECs, suggesting the requirement for bacterial components to their invasion [40]. Moreover, glucose inhibition assay shows that lectin-like interactions are involved in the attachment of *F. nucleatum* to OSCC cells [57]. Furthermore, infection of *F. nucleatum* in human epithelial cells promotes cellular migration, possibly via stimulation of Etk/BMX, S6 kinase p70, and RhoA kinase and increases the production of MMP-13 (collagenase 3) via the activation of mitogen-activated protein kinase p38 [58]. However, the detailed mechanisms of adhesion of *Fusobacterium* in OSCC cells and *F. nucleatum*-mediated invasion of OSCC cells is still unclear. Nevertheless, the fact that *F. nucleatum* is abundant in the oral cavity of OSCC patients seems to be important in the tumorigenesis and/or the progression of oral cancer.

Several reports show the involvement of *F. nucleatum* in tumorigenesis and development of OSCC. Gallimidi et al. demonstrated that chronic intake of *P. gingivalis* and *F. nucleatum* promoted tumor progression in a 4-nitroquinoline-1-oxide (4NQO)-induced mouse tongue cancer model [59]. The infected group showed larger and more invasive tumors with increased expression of cyclin D1, IL-6, and phospho-STAT3. Moreover, co-culture with *P. gingivalis* and *F. nucleatum* upregulated IL-6 expression in OSCC cells, whereas these effects was not caused by *F. nucleatum* alone [59]. These findings suggest that exposure of oral epithelial cells to *P. gingivalis*/*F. nucleatum* triggers TLR signaling, resulting in IL-6 production that activates STAT3 which in turn induces important effectors driving growth and invasiveness of OSCC cells, such as cyclin D1.

It is known that epithelial-to-mesenchymal transition (EMT) in cancer cells is associated with invasion, metastasis, stemness, and resistance of therapy [60]. Recently, *F. nucleatum* promotes cell migration and EMT by upregulating mesenchymal markers, including N-cadherin, Vimentin, and SNAI1 in noncancerous human immortalized oral epithelial cells and OSCC cell lines [61]. As EMT is important event of tumor progression and malignant behaviors, it is worth investigating whether *F. nucleatum* is involved in EMT in OSCC cells. Recently, it has been reported that partial-EMT (p-EMT) that partially combines epithelial and mesenchymal features, is involved in tumor progression and metastasis [62,63]. We previously reported that *SERPINE1*, *ITGA5*, *TGFBI*, *P4HA2*, *CDH13*, and *LAMC2* can be a prognostic marker among p-EMT genes [64]. In our preliminary finding, co-culture with *F. nucleatum* promoted invasion of OSCC cells with upregulation of p-EMT genes [65]. We are now investigating the detailed mechanism of *F. nucleatum*–driven p-EMT in OSCC.

Inflammasomes are multiprotein complexes that regulate immune processes in response to infections and tissue damage. Inflammasomes mediate the processing of the two most important inflammatory cytokines, pro-interleukin-1β (IL-1β), and pro-IL-18 to their active forms. The inflammasome is formed by the apoptosis-associated speck-like protein containing a CARD (ASC), procaspase-1, and a sensor protein, which is either a NOD-like receptor (NLR) or an absent in melanoma 2 (AIM2)-like receptor. IL-1β plays a pro-tumorigenic role and enhances the aggressiveness of OSCC [66]. Aral et al. reported that *F. nucleatum* infection in OSCC cells promotes AIM2 inflammasome expression [67]. In addition, *F. nucleatum* potentially upregulated IL-1β expression, but downregulated POP1 which controls NLRP3 inflammasome activation by targeting ASC. Especially in the oral cavity, it is reported that the presence of nanoparticles released from dental metal alloy activates NLRP3 inflammasome in human oral keratinocytes, resulting in the development of potentially malignant oral disorders [68]. As TRIM16, which enhances IL-1β production, is also upregulated by *F. nucleatum*, *F. nucleatum* may play a role by dysregulating inflammasomes and their modulators in OSCC.

Bacterial infection influences genomic stability and integrity by causing DNA damage, which increases the possibility of tumor initiation and development. Recently, it has been reported that *F. nucleatum* infection in OSCC cells causes DNA damage via Ku70/p53 pathway [69]. Although Ku70 is involved in the non-homologous end joining (NHEJ) pathway of DNA repair via binding to DNA double-strand break ends, the detailed mechanism of the association between *F. nucleatum* infection and Ku70 is still unclear.

It has been shown that there are specific associations among bacteria within dental plaque by cluster analysis using subgingival plaque samples [70,71]. As shown in Figure 1, various bacteria exist as shown in the pyramid. *F. nucleatum* is in the middle of pyramid. Among the species in the middle of pyramid, *F. nucleatum* is dominant in the dental biofilm at a later stage of plaque formation. As described above, *Fusobacterium* infection affects the tumorigenesis and development of OSCC through various responses (Figure 1). However, the target molecules of *Fusobacterium* are still unknown. Further studies will be required for clarifying the evidence of *Fusobacterium* involvement in tumorigenesis and development of OSCC.

## 4. Conclusions

*F. nucleatum* is well known oral commensal bacterium that forms typical dental plaque on human teeth is involved periodontal diseases. The involvement of *F. nucleatum* in carcinogenesis and the development of CRC has attracted attention in the field of CRC research. Importantly, the enrichment of *F. nucleatum* is frequently observed in OSCC tissues compared to healthy oral mucosal tissues. However, the roles of *F. nucleatum* in OSCC are still unclear. It is also necessary to investigate whether the mechanism of *F. nucleatum* in OSCC overlaps with that in CRC. We suggest that oral hygiene managements for reducing the amount of *F. nucleatum* may contribute to the prevention of OSCC and CRC. Moreover, various commensal bacteria exist in the oral cavity. Therefore, it is interesting to examine the relationship between *F. nucleatum* and other bacteria in the tumorigenesis and development of OSCC.

## Figures and Tables

**Figure 1 ijms-21-06207-f001:**
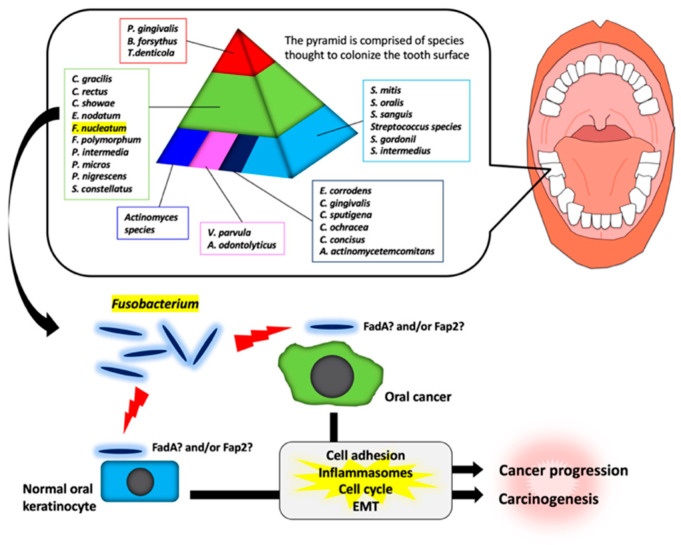
Schematic model for the involvement of *Fusobacterium* species in oral carcinogenesis and cancer progression. In a dental plaque, various bacteria exist as shown in the pyramid. The base of the pyramid is comprised of species thought to colonize the tooth surface and proliferate at an early stage. Then, complex species in the middle of pyramid becomes numerically more dominant later and is thought to bridge the early colonizers. Finally, the complex species in the top of the pyramid numerically more dominant at late stages in plaque development. Among them, *Fusobacterium* species may adhere with oral keratinocyte or oral cancer cells via interaction between FadA/Fap2 and E-cadherin. This interaction may induce carcinogenesis and cancer progression.

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
