# Peer review of "Involvement of Fusobacterium Species in Oral Cancer Progression: A Literature Review Including Other Types of Cancer"

_ijms, 2020, doi:10.3390/ijms21176207_

Round 1

Reviewer 1 Report

TITLE: please indicate the type of study

ABSTRACT: 

page 2, line 22: "chronic inflamation" instead of "inflamation"

In the abstract you should report also the results of your study, not only the literature background.

INTRODUCTION:

page 3, lines 45-57 can be removed, as they do not address the topic of the study

page 3, line 58: add "chronic". Please add a reference to this sentence; as a suggestion, please read the reference below:

Mohammed H, Varoni EM, Cochis A, et al. Oral Dysbiosis in Pancreatic Cancer and Liver Cirrhosis: A Review of the Literature. Biomedicines. 2018;6(4):115. Published 2018 Dec 11. doi:10.3390/biomedicines6040115

page 4, lines 77-79: please add a reference to this sentence; as a suggestion, please read the reference below:

Patini R, Staderini E, Lajolo C, et al. Relationship between oral microbiota and periodontal disease: a systematic review. Eur Rev Med Pharmacol Sci. 2018;22(18):5775-5788. doi:10.26355/eurrev_201809_15903

page 4, lines 83-84: please rephrase this sentence

page 7, line 6: "two type of aherence..." is it something missing? "Adherence proteins"page 7, lines 142-143: there are two "recognize", please rephrase this sentence.

page 7, line 144: please check the verb tense of "localizes"

page 8, line 159: "were" instead of "was"

page 8, line 162: the verb tense should be uniformed; in the previous sentence it is used the past simple, while in this sentence it is used the present simple.

page 9, line 189: "of adhesion" instead "on adhesion"

page 9, lines 195-196: the verb thenses are not consistent!

page 10, line 198: the "increased expression of cyclin D1, IL-6 and phospho-STAT3" has a clinical meaning on cancer prognosis? 

MATERIALS AND METHODS: missing

RESULTS: missing

DISCUSSION: missing

CONCLUSION: missing

REFERENCES:

Author Response

This reviewer asks that we address the following specific issues. We appreciate that the referee is enthusiastic about certain aspects. He/she asks that we address the following specific issues.

  • TITLE: please indicate the type of study.  We changed the title as “Involvement of Fusobacterium species in oral cancer progression: A literature Review including other types of cancer”.
  • ABSTRACT:page 2, line 22: "chronic inflamation" instead of "inflamation".   We changed as suggested by this reviewer.
  • In the abstract you should report also the results of your study, not only the literature background.  Our findings are preliminary. Therefore, we cannot add the results of our study in the "Abstract”. However, we added our preliminary findings in the main text as the following; Recently, it has been reported that partial-EMT (p-EMT) that partially combines epithelial and mesenchymal features, is involved in tumor progression and metastasis [62, 63]. We previously reported that SERPINE1, ITGA5, TGFBI, P4HA2, CDH13, and LAMC2 can be a prognostic marker among p-EMT genes [64]. In our preliminary finding, co-culture with F. nucleatum promoted invasion of OSCC cells with upregulation of p-EMT genes (Fujiwara et al., unpublished data). We are now investigating the detailed mechanism of F. nucleatum-driven pEMT in OSCC.
  • INTRODUCTION: page 3, lines 45-57 can be removed, as they do not address the topic of the study.  As suggested by this reviewer, we deleted the sentences (lines45-57).
  • page 3, line 58: add "chronic". Please add a reference to this sentence; as a suggestion, please read the reference below: Mohammed H, Varoni EM, Cochis A, et al. Oral Dysbiosis in Pancreatic Cancer and Liver Cirrhosis: A Review of the Literature. Biomedicines. 2018;6(4):115. Published 2018 Dec 11. doi:10.3390/biomedicines6040115.  We added “Chronic”. Moreover, we also added a reference as suggested by this reviewer.
  • page 4, lines 77-79: please add a reference to this sentence; as a suggestion, please read the reference below: Patini R, Staderini E, Lajolo C, et al. Relationship between oral microbiota and periodontal disease: a systematic review. Eur Rev Med Pharmacol Sci. 2018;22(18):5775-5788. doi:10.26355/eurrev_201809_15903.  We added a reference as suggested by this reviewer.
  • page 4, lines 83-84: please rephrase this sentence.  We rephrased as “F. nucleatum has two types of substance, fimbriae and non-fimbrial adhesin, for attaching to the other bacteria and cells”.
  • page 7, line 6: "two type of adherence..." is it something missing? "Adherence proteins".  We rephrased as “Two distinct types of adherence protein have been found in fusobacteria and classified based on their inhibition mode using either D-galactose or L-arginine”.
  • page 7, lines 142-143: there are two "recognize", please rephrase this sentence.  We changed the sentence, “Moreover, acetylgalactosamine (Gal-GalNAc) is recognized by fusobacterial Fap2, which functions as a Gal-Gal-NAc lectin”.
  • page 7, line 144: please check the verb tense of "localizes"We changed as pointed out by this reviewer.
  • page 8, line 159: "were" instead of "was".  We changed as pointed out by this reviewer.
  • page 8, line 162: the verb tense should be uniformed; in the previous sentence it is used the past simple, while in this sentence it is used the present simple.  Thank you very much for pointing out. We uniformed the verb tense as pointed out.
  • page 9, line 189: "of adhesion" instead "on adhesion".  We changed as suggested.
  • page 9, lines 195-196: the verb senses are not consistent!  We changed as pointed out by this reviewer.
  • page 10, line 198: the "increased expression of cyclin D1, IL-6 and phospho-STAT3" has a clinical meaning on cancer prognosis?  Increased expression of cyclin D1, IL-6 and phospho-STAT3 may be involved in growth and invasiveness of OSCC cells. Therefore, we added the following sentence;“These findings suggest that exposure of oral epithelial cells to P. gingivalis/F. nucleatum triggers TLR signaling, resulting in IL-6 production that activates STAT3 which in turn induces important effectors driving growth and invasiveness of OSCC cells, such as cyclin D1.”

Reviewer 2 Report

The review article written by Fujiwara et al. discussed the link between Fusobacterium species and oral, colorectal, and esophageal cancers progression. This manuscript presents significant issue, however, I have some following minor as well as major points.

Major points

First of all, the title, i.e.Fusobacterium species Associated with Oral Cancer Progression” does not reflect the content of article. Besides oral cancer, the authors described colon and esophageal cancers. Therefore, it is recommended to alter the title.

Moreover, the “introduction” section should be reorganized. There should also be included the risk factors for colon as well as esophageal cancers (if authors present the risk factors for oral cancer). This section may be developed using the following reference (in this paper the risk factors for colorectal as well as oral cancers are described precisely):

Kaźmierczak-Siedlecka, K.; DvoÅ™ák, A.; Folwarski, M.; Daca, A.; PrzewÅ‚ócka, K.; Makarewicz, W. Fungal Gut Microbiota Dysbiosis and Its Role in Colorectal, Oral, and Pancreatic Carcinogenesis. Cancers 202012, 1326.

Additionally, the authors should explain why they have chosen this type of cancers. Is it because Fusobacterium species take part in progression of only oral, colon, and esophageal cancers?

The role of Fusobacterium nucleatum in development of colorectal cancer also needs further development. The authors can use following articles:

  1. Fusobacterium nucleatum promotes colorectal cancer metastasis by modulating KRT7-AS/KRT7”

(https://www.tandfonline.com/doi/full/10.1080/19490976.2019.1695494)

  1. “Therapeutic methods of gut microbiota modification in colorectal cancer management – fecal microbiota transplantation, prebiotics, probiotics, and synbiotics”

(https://www.tandfonline.com/doi/full/10.1080/19490976.2020.1764309)

The section “Fusobacterium species and colorectal and esophageal cancers” should be divided into two different parts. It is strongly needed to describe colorectal cancer followed by esophageal cancer. The data regarding the link between these bacterial species and development of esophageal cancer is strongly limited and it should be emphasized in this article.

Moreover, there is a lack of “conclusions” section. It is necessary to provide precisely described summary of this paper presenting future directions and possibilities of using of these Fusobacterium species in clinical practice.

Minor points

It is recommended to read whole manuscript line by line and correct language and typo errors.

For instance: lines 23, 24, 52, 65 …

Line 63 – word “species” should not be written in italics.

Overall, this paper is worth to be published, nevertheless, the alterations are necessary.

Author Response

This Reviewer is enthusiastic about our results stating that: “This manuscript presents significant issue”. He/she asks that we address the following specific issues.

Major points

  • First of all, the title, i.e. “Fusobacterium species Associated with Oral Cancer Progression” does not reflect the content of article. Besides oral cancer, the authors described colon and esophageal cancers. Therefore, it is recommended to alter the titleAs suggested by this reviewer, we changed the title as “Involvement of Fusobacterium species in oral cancer progression: A literature Review including other types of cancer”.
  • Moreover, the “introduction” section should be reorganized. There should also be included the risk factors for colon as well as esophageal cancers (if authors present the risk factors for oral cancer). This section may be developed using the following reference (in this paper the risk factors for colorectal as well as oral cancers are described precisely): Kaźmierczak-Siedlecka, K.; DvoÅ™ák, A.; Folwarski, M.; Daca, A.; PrzewÅ‚ócka, K.; Makarewicz, W. Fungal Gut Microbiota Dysbiosis and Its Role in Colorectal, Oral, and Pancreatic Carcinogenesis. Cancers 2020, 12, 1326.  We changed the “introduction” section as suggested by this reviewer. We also added the reference.
  • Additionally, the authors should explain why they have chosen this type of cancers. Is it because Fusobacterium species take part in progression of only oral, colon, and esophageal cancers?  So far, it has been reported the association with F. nucleatum and colorectal/esophageal cancer. However, F. nucleatum is well known as ubiquitous and pathogenic bacteria in the oral cavity. Therefore, we thought that Fusobacterium species may be involved in the development of OSCC. Indeed, Fusobacterium was frequently detected in OSCC tissues, compared with normal oral mucosal tissues. We added this point in “Introduction”.
  • The role of Fusobacterium nucleatum in development of colorectal cancer also needs further development. The authors can use following articles: 1) “Fusobacterium nucleatum promotes colorectal cancer metastasis by modulating KRT7-AS/KRT7”.  2) “Therapeutic methods of gut microbiota modification in colorectal cancer management – fecal microbiota transplantation, prebiotics, probiotics, and synbiotics” .  As suggested by this reviewer, we added sentences showing the role of Fusobacterium nucleatum in development of colorectal cancer by using two articles.
  • The section “Fusobacterium species and colorectal and esophageal cancers” should be divided into two different parts. It is strongly needed to describe colorectal cancer followed by esophageal cancer. The data regarding the link between these bacterial species and development of esophageal cancer is strongly limited and it should be emphasized in this article.  Thank you for suggestion. We divided two different parts as the reviewer’s suggestion.
  • Moreover, there is a lack of “conclusions” section. It is necessary to provide precisely described summary of this paper presenting future directions and possibilities of using of these Fusobacterium species in clinical practice.  We provided “Conclusion” section included the sentences of summary of this paper presenting future directions and possibilities of using of these Fusobacterium species in clinical practice.         

Minor points

  • It is recommended to read whole manuscript line by line and correct language and typo errors. For instance: lines 23, 24, 52, 65 …  Thank you very much for pointing out. As much as possible, we corrected language and typo errors.
  • Line 63 – word “species” should not be written in italicsWe changed as pointed out by this reviewer.
  • Overall, this paper is worth to be published, nevertheless, the alterations are necessary.  Thank you very much for your comments.

Reviewer 3 Report

 In this review, authors introduce the reports focused on the association of Fusobacterium species with cancer development and progression including oral, esophageal, and colon cancers. The review provides an very important contribution to Fusobacterium species with cancer development and progression including oral, esophageal, and colon cancers.

However, minor revision of manuscript is needed before it can be accepted for publication.

1)With regard to content, there appears to be a lack of introduction section. I would anticipate that adding this more complete introduction would also require the addition of a number of additional references.

2) The conclusion part is short and to the point, but again does not clearly show the importance and relevance of the review’s findings.

Author Response

This Reviewer stated that “The review provides a very important contribution to Fusobacterium species with cancer development and progression including oral, esophageal, and colon cancers.”  He/she asks that we address the following specific issues.

  • With regard to content, there appears to be a lack of introduction section. I would anticipate that adding this more complete introduction would also require the addition of a number of additional referencesAs other two reviewers pointed out, we added some sentences and several references in “Introduction” section.
  • The conclusion part is short and to the point, but again does not clearly show the importance and relevance of the review’s findingsAs suggested by this reviewer, we provided “Conclusion” section.

Round 2

Reviewer 1 Report

LANGUAGE: there are several typos. I think that this literature review can be considered for publication only after a professional language editing.

TITLE: "Review" should be written without capital letters. Moreover, I would suggest to remove "including other thypes of cancer"

ABSTRACT: 

page 1, line 21: you can use: "it has recently been shown" or "Recently, it has been shown". You have to remove "that", otherwise you need to add a verb.

page 1, line 22-23: "As over 700...cancer" I would remove this sentence

page 1. lines 23-26: I would rephrase this sentence, e.g.: "Despite Several papers have shown that Fusobacterium species play a crucial role for the development of colorectal and esophageal cancer, it remains unclear the role of bacteria regarding tumorigenesis of oral cancer"

Even if the results are still preliminary, you need to summarize the preliminary information in the results, discussion and conclusion of the abstract.

INTRODUCTION:

page 2, line 87: please add a reference dealing with oral infections. As a suggestion, please find a reference below:

Staderini E, Patini R, Guglielmi F, Camodeca A, Gallenzi P. How to Manage Impacted Third Molars: Germectomy or Delayed Removal? A Systematic Literature Review. Medicina (Kaunas). 2019;55(3):79. Published 2019 Mar 26. doi:10.3390/medicina55030079